**Typical meteorological conditions associated with extreme nitrogen dioxide**
**(NO₂) pollution events over Scandinavia**
Manu Anna Thomas and Abhay Devasthale
Research and development department, Swedish Meteorological and Hydrological Institute
(SMHI), Folkborgsvägen 17, 60176 Norrköping, Sweden
Correspondence: manu.thomas@smhi.se
**Abstract**
Characterizing typical meteorological conditions associated with extreme pollution events helps in
the better understanding of the role of local meteorology in governing the transport and distribution
of pollutants in the atmosphere. The knowledge of their co-variability could further help to evaluate
and constrain chemistry transport models (CTMs). Hence, in this study, we investigate the statistical
linkages between extreme nitrogen dioxide (NO₂) pollution events and meteorology over
Scandinavia using observational and reanalysis data. It is observed that the south-westerly winds
dominated during extreme events, accounting for 50-65% of the total events depending on the
season, while the second largest annual occurrence was from south-easterly winds, accounting for
17% of total events. The specific humidity anomalies showed an influx of warmer and moisture-
laden air masses over Scandinavia in the free troposphere. Two distinct modes in the persistency of
circulation patterns are observed. The first mode lasts for 1-2 days, dominated by south-easterly
winds that prevailed during 78% of total extreme events in that mode, while the second mode lasted
for 3-5 days, dominated by south-westerly winds that prevailed during 86% of the events. The
combined analysis of circulation patterns, their persistency, and associated changes in humidity and
clouds suggests that NO₂ extreme events over Scandinavia occur mainly due to the long-range
transport from the southern latitudes.





## 1. Introduction

Nitrogen dioxide ($NO_2$) is one of the highly reactive gases of the nitrogen oxides (NOx) family. The major sources of $NO_2$ are fuel combustion in motor vehicles, industrial boilers, emissions from soil and agricultural biomass burning. The natural source of $NO_2$ is lightning and forest fires. Recent studies indicate increasing trends in $NO_2$ in developing countries and decreasing trends in developed countries as a result of environmental regulation policies (Richter et al. 2005; Zhang et al. 2007; van der A et al. 2008; Schneider et al. 2015; Geddes et al. 2016). $NO_2$ is an oxidizing agent resulting in the corrosive nitric acid and plays an important role aiding the formation of ozone. It can also contribute to the formation of particulate matter (PM) and secondary organic particles through photochemical reactions. Increased NOx concentrations not only severely affect human physical health through reduced lung function, but also affect aquatic ecosystems through acid deposition and eutrophication of soil and water (Sjöberg et al. 2004; Klingberg et al. 2009; Bellandar et al. 2012; Gustafsson et al. 2014; Nilsson Sommar et al. 2014; Oudin et al. 2016; Taj et al. 2016). Lamarque et al. (2013) based on the multi-model intercomparison assessed increases in regional nitrogen deposition by up to 30-50% from RCP 2.6 to RCP 8.5. According to the 4th IPCC Assessment Report, the total global NOx emissions have increased from a pre-industrial value of 12 Tg N/yr to between 42 and 47 Tg N/yr in 2000. The most recent study by Miyazaki et al. (2017) estimated a ten year (2005-2014) global total surface NOx emissions of 48.4 Tg N/year with an increase of 29%, 26% and 20% per decade increase respectively over India, China and Middle East and a decrease of 38%, 8.2% and 8.8% respectively over United States, southern Africa and western Europe. In heavily polluted areas $NO_2$ can also have noticeable impact on the local radiation budget (Vasilkov et al. 2009).

Compared to other pollutants such as carbon monoxide (CO) that has a life span of weeks to few months, $NO_2$ has a relatively shorter life time in the atmosphere and ranges typically from a couple of hours in the boundary layer to up to few days in the upper troposphere (Beirle et al., 2011). Therefore, $NO_2$ can be typically associated with short-range transport events. For long range transport (LRT) or intercontinental transport of pollutants and in particular of $NO_2$ to occur, the associated weather systems need to be linked with stronger winds and rapid convective-advective events such as cyclones or warm conveyor belts (WCBs) that can lift air masses from their source regions up into the free troposphere and be transported across the oceans (Eckhardt et al. 2003; Stohl et al. 2003). Due to lower concentrations of radical species in the free troposphere, the reaction with $NO_2$ is limited. Zien et al. (2014) identified about 3800 LRT events of $NO_2$ during a 5

year period from the major pollution hotspots such as the east coast of North America, central
Europe, China and South America, predominantly during autumn and winter months.

There have been several studies reporting individual LRT events of $NO_2$. To mention a few, Stohl et
al. (2003) in a study explained "intercontinental express highways" being responsible for almost
60% of the total intercontinental transport of pollutants from across the Atlantic to Europe, resulting
in an increment of average European winter NOx mixing ratios by about 2-3 pptv. In yet another
study, Schaub et al. (2005) demonstrated that at least 50 % of the $NO_2$ recorded at the Alpine region
was advected via a frontal system from the Ruhr area in central Germany in February 2001.
Donnelly et al. (2015) reported that easterly air masses during winter resulted in increased $NO_2$
concentrations in the urban and rural sites in Ireland. LRT of NOx across the Indian Ocean from
South Africa to Australia in May 1998 was reported by Wenig et al. (2003).

The Nordic countries often lie at the receiving end of short-range pollutant transport from northern
Europe or they are a part of a much larger transit pathway of eventual long-range transport to the
Arctic, originating from either Europe or North America. To what extent such a transport from the
southerly latitudes affects the characteristics of extreme pollution events (such as magnitude,
frequency and persistence) over Scandinavia depends largely on prevailing circulation patterns and
meteorological conditions. The local meteorology can enhance or dampen the concentration of the
pollutants depending on the degree of persistency; the knowledge of which would help to better
constrain the chemistry transport models (CTMs). Therefore, identifying the dominant weather
patterns over Scandinavia especially during extreme pollution is important. However, there has not
been a systematic study linking the transport events of $NO_2$ to different meteorological conditions,
solely from observational data over the Scandinavian region. Therefore, the main aim of the present
study is to characterize circulation regimes and meteorological conditions extreme pollution events,
to understand to what extent they differ from climatological conditions. There are two different
ways to study this co-variability solely using observational data: 1) the "top- down approach"
wherein the atmospheric state is first identified and then the variability of the tracers is evaluated.
This approach gives a general perspective of the distribution of tracers based on a particular weather
state and 2) the "bottom-up approach" wherein the pollution episode is first identified and the
weather state associated with it is studied. In this study we make use of the bottom-up approach as
explained in the next section.



## 2. Data sets and methodology

The NO$_2$ tropospheric column densities from OMI (Ozone Monitoring Instrument) on board the EOS Aura satellite are used in this study to define and identify extreme events (Boersma et al., 2001, 2008, 2011; Bucsela et al., 2006, 2008, 2013; Lamsal et al. 2008, 2010, 2014). 11 years (2004 – 2015) of daily Level 3 gridded standard product, available at 0.25x0.25 degrees resolution is analysed (OMNO2d, Version 3, available at: https://disc.gsfc.nasa.gov/Aura/data-holdings/OMI/omno2d_v003.shtml). This particular product is used as it provides good quality OMI retrievals, already screened based on recommendations by the OMI Algorithm Team. We allowed retrievals under cloudy conditions to be analysed, not only to have robust number of samples, but also to avoid clear-sky biases since the NO$_2$ transport is often associated with cyclonic systems that lead to increased cloudiness (Zien et al. 2014). We further tested the sensitivity of our results to using only cloud screened retrievals, to evaluate if the selection of extreme events and associated meteorological conditions are different from those cases when retrievals under partially cloudy conditions are used.

Humidity and cloud fraction retrievals from the AIRS (Atmospheric Infrared sounder) instrument on board Aqua satellite are used (Chahine et al. 2006; Susskind et al. 2014; Devasthale et al. 2016). Both Aqua and Aura satellites are a part of NASA's A-Train convoy, providing added advantage of simultaneous observations of trace gases from OMI-Aura and thermodynamical information from AIRS-Aqua. AIRS Version 6 Standard Level 3 Daily Product (AIRX3STD) for the same period (2004-2015) is used (data available at: https://disc.gsfc.nasa.gov/uui/datasets?keywords=%22AIRS%22).

To investigate circulation patterns, u and v wind components at 850 hPa from ECMWF's ERA-Interim Reanalysis are used (Dee et al., 2011; http://apps.ecmwf.int/datasets/data/interim-full-daily/levtype=sfc/).

In order to investigate co-variability of meteorological conditions and pollutants using observations, two different approaches can be taken (Fig. 1). In a "top-down" approach, a weather state classification can be done to identify most prevailing weather states that occur over the study area and then the relative distribution of pollutants can be investigated under those states to rank them. This approach was adapted by Thomas and Devasthale (2014) and Devasthale and Thomas (2012). In a "bottom-up" approach on the other hand, a set of pollution events can be identified first and then the corresponding meteorological conditions can be investigated. This bottom-up approach is

the focus of the present study. It should be mentioned that both of these approaches have their

advantages and limitations. For example, the dominant weather pattern identified in the top-down

approach may not have the largest impact on pollutant variability and the pollution events identified

in the bottom-up approach may not be associated with the dominant weather pattern or may not

have the largest impact on an average in the weather state they occur. Therefore, only the

combination of these two approaches will provide a complete picture of the co-variability between

meteorological conditions and pollutants.

In the present study, an "extreme" pollution event is defined as follows. First, the histograms of

$NO_2$ tropospheric column densities using OMI data for each month are computed over the centre of

the study area (55N-60N, 11E-20E). This area is chosen because it accommodates top ten polluted

and populated cities/regions in Sweden (Sjöberg et al. 2004; Klingberg et al. 2009; Bellandar et al.

2012; Gustafsson et al. 2014; Nilsson Sommar et al. 2014; Oudin et al. 2016; Taj et al. 2016 ). All

events that surpass the 90-percentile (90%ile) value are considered as extreme events. The monthly

histograms of $NO_2$ over the study region are shown in Fig. 2 along with 90 percentile thresholds for

each month (vertical lines). Since $NO_2$ distributions over the study area show strong monthly

variability, the monthly thresholds were chosen to define extreme events.  The distributions of $NO_2$

have longer tails during winter half year and the tropospheric columns are also higher. Therefore,

the resulting 90%ile thresholds are also higher in winter compared to summer months. However,

using thresholds based on percentiles (rather than having a fixed value throughout the season or

year), makes the criteria for the selection of extreme events fair and equally applicable for each

month.

**3. Meteorological conditions observed during extreme events**

The spatial distribution of tropospheric $NO_2$ column during climatological conditions, extreme

events and anomalies thereof is presented in Fig. 3. Note that although the thresholds for defining

extreme events are different for each month, the results are compiled over four distinct seasons for

the sake of brevity. By definition, $NO_2$ anomalies during extreme events are similar in magnitude to

climatological values over Scandinavia. The spatial extent of the severity of the extreme pollutant

episodes over southern Sweden is noticeable. Under climatological conditions, highest

concentrations are observed over northern Germany and France, the Netherlands and Belgium (the

Benelux region). There is a good spatial coherence between $NO_2$ distributions under climatological

conditions and extreme events, in the sense that the high concentrations of $NO_2$ seemed to have

spread over southern Scandinavia during extreme events from the regions where climatological

values are usually higher. It is to be noted that during extreme events the pollution levels over
northern European regions are also enhanced. For an event to qualify as an extreme event over
southern Scandinavia, the pollutant levels in the source regions also need to be higher than usual in
order to allow strong transport under favourable atmospheric circulation patterns. This provides
confidence in the selection process of extreme events. The $NO_2$ concentrations are relatively higher
in winter and autumn compared to the summer months. This is mainly because atmospheric
removal by radical species and deposition are much more efficient in the summer months.

In order to characterize typical meteorological conditions that can result in such high concentrations
over Scandinavia, we first investigated the dominant wind direction at 850 hPa associated with
those extreme events using ERA-Interim reanalysis data.  The normalized frequency of occurrence
of different wind directions during four seasons is shown in Fig. 4. It can be seen that, irrespective
of the season, the south-westerly winds are dominant during extreme events accounting for 50-65%
of total events. This is consistent with south-westerly extension of pollution plume mentioned
earlier. The second largest annual occurrence is from south-easterly winds, accounting for 17% of
total events followed equally similar contribution from north-westerly winds. Compared to
climatological conditions, south-westerly winds have 30-40% more likelihood of being dominant
during extreme events depending on the season. However, such clear tendency compared to
climatological conditions is not observed in the case of other wind directions. The spatial pattern of
the 850 hPa winds based on ERA-Interim reanalysis and corresponding humidity anomalies at 850
hPa based on AIRS data during extreme events are shown in Figs. 5 and 6 respectively. A clear
transport pathway from the northern continental Europe to Scandinavia is visible. The strongest
winds are observed during the DJF months followed by the SON months with average wind speeds
reaching over 10 m/s. The weakest winds are observed during the JJA months. The circulation
pattern is characterized by the presence of low pressure systems in the Norwegian Sea that create
favourable conditions for the transport of pollutants from continental Europe into Scandinavia. The
location of the center of these cyclonic systems can slightly vary over the Norwegian Sea, affecting
the direction and strength of the northward flow, as evident in Fig. 5.  For example, in the DJF
months, the center is located far away in the open Norwegian Sea allowing stronger south-westerly
winds over southern Scandinavia. In the JJA months, the center of cyclonic systems is close to
western Norwegian coast. While this pattern also leads to south-westerly winds, air masses are
mixed with colder and drier air from the northern Norwegian Sea.

The specific humidity anomalies show an influx of warmer and moister air masses over Scandinavia
(Fig. 6), except in summer as mentioned above. The seasonality in the vertical structure of the
specific humidity anomalies over Scandinavia is shown in Fig. 7c. While there are large deviations
in humidity anomalies, influenced by the strength of the wind flow, they are positive regardless of
the season during extreme events and peak at 2-3 km above the surface.  Such increase in the free
tropospheric moisture, especially during winter half year in the absence of local moisture sources,
can only be explained by the transport from southern latitudes. The vertical water vapour anomalies
are higher in winter half year (DJF and SON), consistent with high $NO_2$ anomalies during those
months. Fig. 8 further shows cloud fraction anomalies. Average cloudiness is increased in all
seasons during extreme events, in particular during winter half year.  During this time of year, the
large-scale frontal systems originating from the southwesterly regions can bring moister airmasses
over Scandinavia, as can be seen in the circulation patterns and humidity anomalies, creating
favourable conditions for cloud formation. Therefore, these positive cloud fraction anomalies, in
combination with positive humidity anomalies and circulation patterns, are indicative of the long-
range transport of airmasses associated with increased $NO_2$ concentrations.

For an extreme pollution event to be linked with the transport the wind flow should be stronger
allowing rapid advection and associated circulation pattern also needs to be persistent. Fig. 7a and
7b show the histograms of wind speed at 850 hPa over the study areas during extreme events when
data are partitioned by wind direction and by season respectively. The average values of wind
speeds are also shown for extreme events and climatological conditions (in brackets). Although the
distributions are shifted to higher wind speeds in nearly all cases during extreme events compared
to climatological conditions, the average wind speeds are not significantly different. The south-
westerly winds are strongest and show largest difference in average wind speeds, while the
northeasterly winds are weakest. Average wind speeds during the winter half year (DJF and SON)
are higher than the summer half year, consistent with observed positive anomalies of humidity and
clouds.

The persistency of the different circulation patterns during these extreme events is further evaluated
as shown in Fig. 7d. The persistency is defined as follows. If an extreme event is observed, the wind
speed and wind direction are computed for the last 10 days. It is then checked how many days back
in time that particular wind direction was *continuously* sustained and that wind direction is not
changed by more than $\pm15^0$ (a third of the quadrant) during that time period. It is to be noted that
the choice of the $\pm15^0$ threshold is based on the visual inspection of about 25 test cases. It was
found that if a stricter threshold is used (requiring wind direction deviations less than $\pm5^0$) the
sampling is considerably reduced for long persistency events. On the other hand, if a more relax
threshold is used (allowing deviations up to $\pm30^0$) we incorporate tail ends of the events that
persisted over neighbouring areas. Two distinct modes in the persistency of circulation patterns are
observed, one in which a particular wind direction persists for a day or two and a second mode in
which winds persists for 3 to 5 continuous days. This is clearly different from the degree of
persistency observed under climatological conditions when winds persisted in one particular
direction predominantly for few days. It was identified that during extreme events south-easterly
winds dominated the first mode explaining 78% of the total occurrence in that mode and the
westerly winds dominated the second mode explaining 86% of the total occurrence. In the latter
case, when the winds persist for few days (3-5 days), the conditions are favourable for the long-
range transport from the southern latitudes since circulation patterns (Fig. 5) are associated with
typical frontal systems and baroclinic disturbances that make their way over Scandinavia.


**4. Sensitivity of chosen events to cloud clearing procedure**

As mentioned in Section 2, we allowed retrievals under cloudy conditions to be analysed, not only
to have a robust number of samples, but also to avoid potential clear-sky biases. However, clouds
can contaminate the $NO_2$ retrievals by modulating scattering in the atmosphere. Moreover, clouds
are highly variable not only in space and time but also in their nature, thus making it challenging to
assess their overall impact on the quality of retrievals. In the case of our study, potential cloud
contamination can affect the selection of extreme events and thereby associated weather patterns
that are being studied. Therefore, we carried out a sensitivity study wherein the entire analysis was
repeated using only cloud screened $NO_2$ retrievals to investigate to what extent cloud clearing
would affect the chosen events and subsequent analysis. We required that cloud fraction is less than
10% in AIRS data and valid retrievals of OMI cloud cleared tropospheric column $NO_2$ are available.
Fig. 9 shows the histograms of $NO_2$ total columns under partially cloudy (solid lines) and cloud
screened conditions (dotted lines). The histograms are accumulated over four seasons instead of
months for clarity (to avoid too many lines). The chosen 90%ile thresholds are certainly different
under partially cloudy and cloud screened conditions, but only slightly. We also found that,
depending on the month, the selected extreme events match under partially cloudy and cloud
screened conditions between 76% and 88% of the time. Fig. 10 further shows the spatial
climatological distribution of $NO_2$ and during extreme events using only cloud screened retrievals.
When compared to Fig. 3, the spatial distributions look patchy as a result of selected screening, but
the magnitude and spatial features do not change significantly, providing confidence in our earlier
analysis based on partially cloudy retrievals. Finally we evaluated if the events based on cloud
screened data impact the analysis of meteorological conditions investigated here. Fig. 11 shows the

vertical structure of specific humidity anomalies over the study region under partially cloudy (solid lines) and cloud screened conditions (dotted lines). While the slight differences in the vertical structure do exist, their sign and magnitudes are not large enough to change any previous argumentation.

**5. Conclusions**

The main aim of the present study was to characterize typical meteorological conditions associated with extreme $NO_2$ pollution events over Scandinavia. To that end, the study employs the bottom-up approach, in contrast to top-down approach taken by Thomas and Devasthale (2014) to study statistical co-variability of weather states and pollutant distribution. Such detailed analysis characterizing circulation patterns and meteorological conditions involving more than 300 extreme pollution events identified using satellite data has not been done before over the Scandinavian region. It is observed that the south-westerly winds dominated during extreme events accounting for 50-65% of total events, while the second largest annual occurrence was from south-easterly winds, accounting for 17% of total events followed by an equally similar contribution from north-westerly winds. Wind speeds are generally higher during extreme events, but only slightly, making it challenging to delineate distinct circulation regimes under these events. For the first time, we investigated the degree of persistency of wind direction during extreme events. In contrast to climatological conditions, two distinct modes of persistency were found; first one lasting a day or so and dominated by winds from south-easterly direction and the other mode lasting 3 to 5 days dominated by south-westerly and north-westerly winds. This information on the degree of persistency in conjunction with circulation patterns could be useful to identify extreme transport events. Further analysis of circulation patterns in combination with spatial distribution of humidity and its vertical structure suggest that these events occur as a result of long-range transport from southern latitudes, most likely from the northern parts of Germany and France, the Netherlands and Belgium. The analysis presented here provides information that can be used in the process oriented evaluation of chemistry transport models over Scandinavia.

**Acknowledgements**

We gratefully acknowledge OMI and AIRS Science Team and NASA GES DISC for providing data. The wind data from ERA-Interim reanalysis have been obtained from the ECMWF Data Server. MT acknowledges funding support from the Swedish Clean Air and climate research program of

IVL (Swedish Environmental Research Institute). Both MT and AD acknowledge Swedish National
Space Board (grants 84/11:1, 84/11:2, Dnr: 94/16).

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

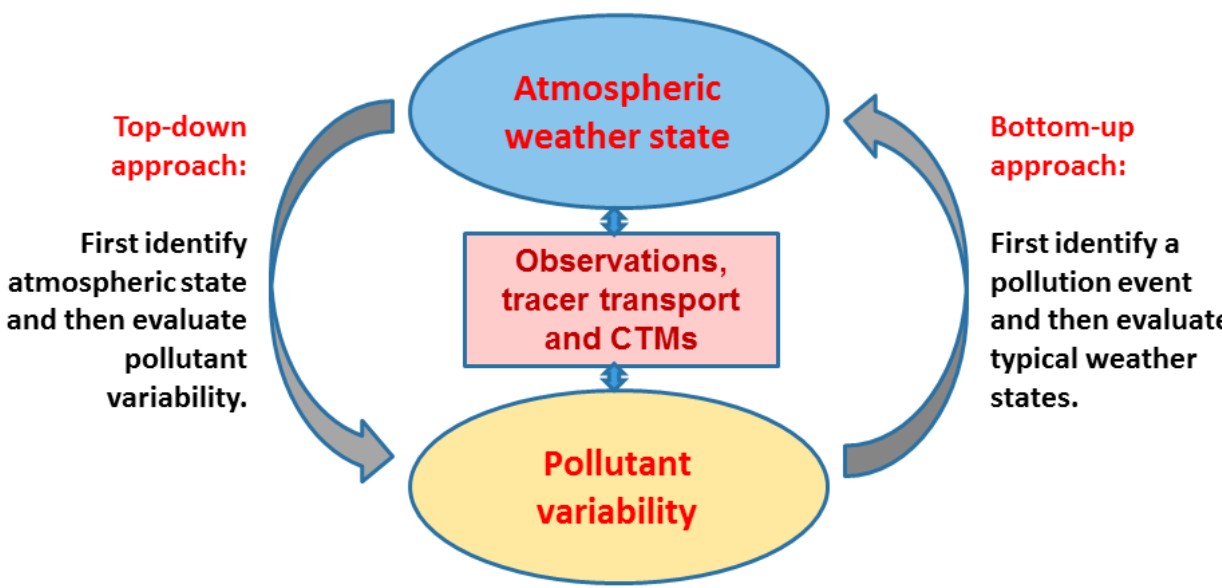


Fig. 1: Schematic showing two different approaches to study statistical co-variability of
atmospheric weather states and pollutant concentrations.














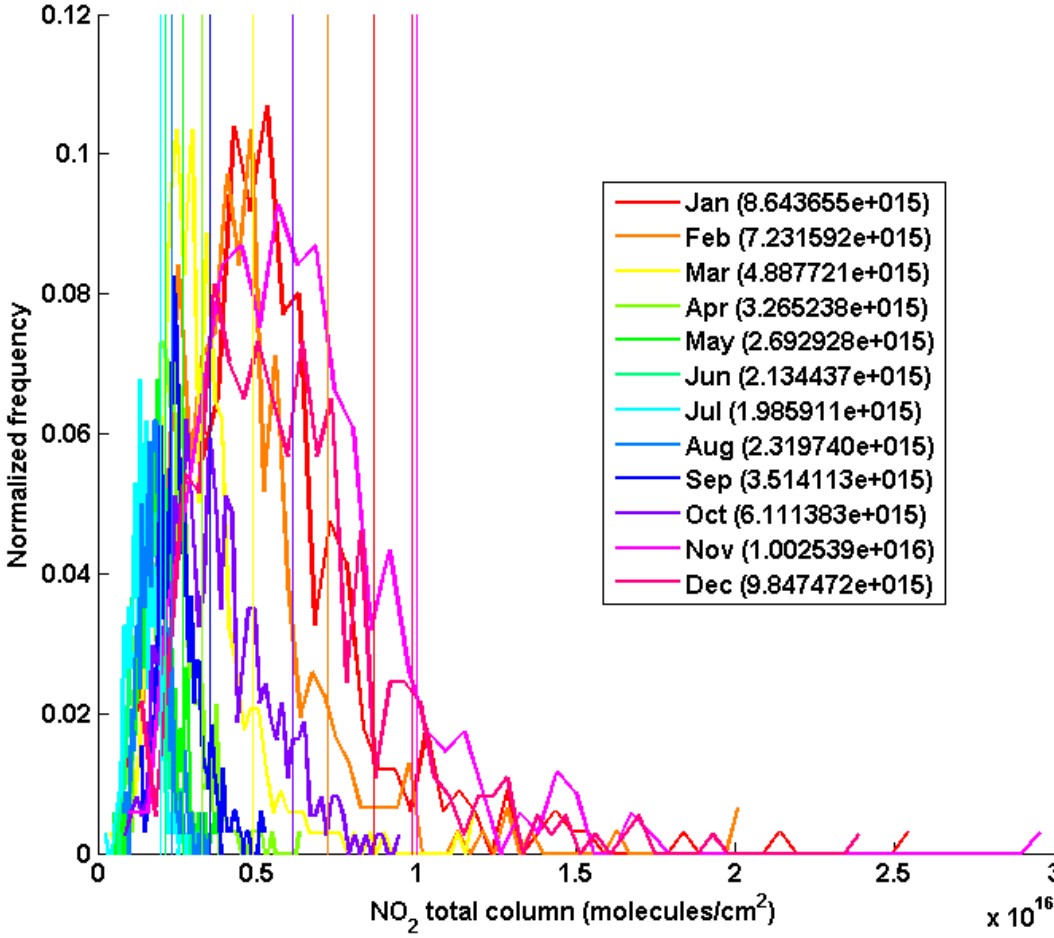

Fig. 2: Monthly histograms of tropospheric total column $NO_2$ over the centre of the study area (55N-60N, 11E-20E) and corresponding 90%ile thresholds (shown by vertical lines and values in brackets).

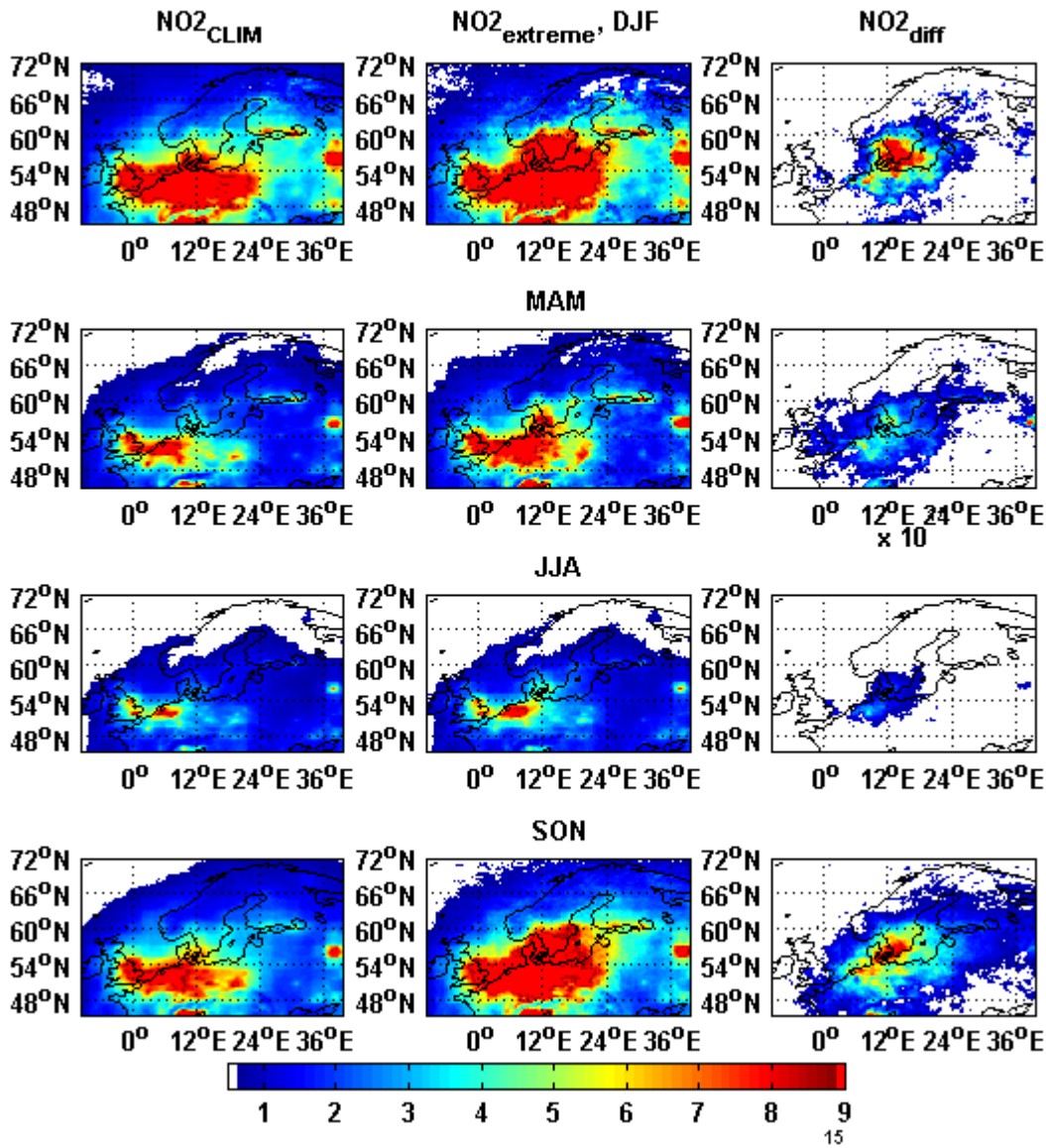

Fig, 3: Seasonal, climatological average tropospheric $NO_2$ total column (first column) based on
nearly 11-yr OMI data (2004-2015), $NO_2$ distribution during extreme events (second column) and
the difference between the two (third column). The units are in molecules/cm$^2$.

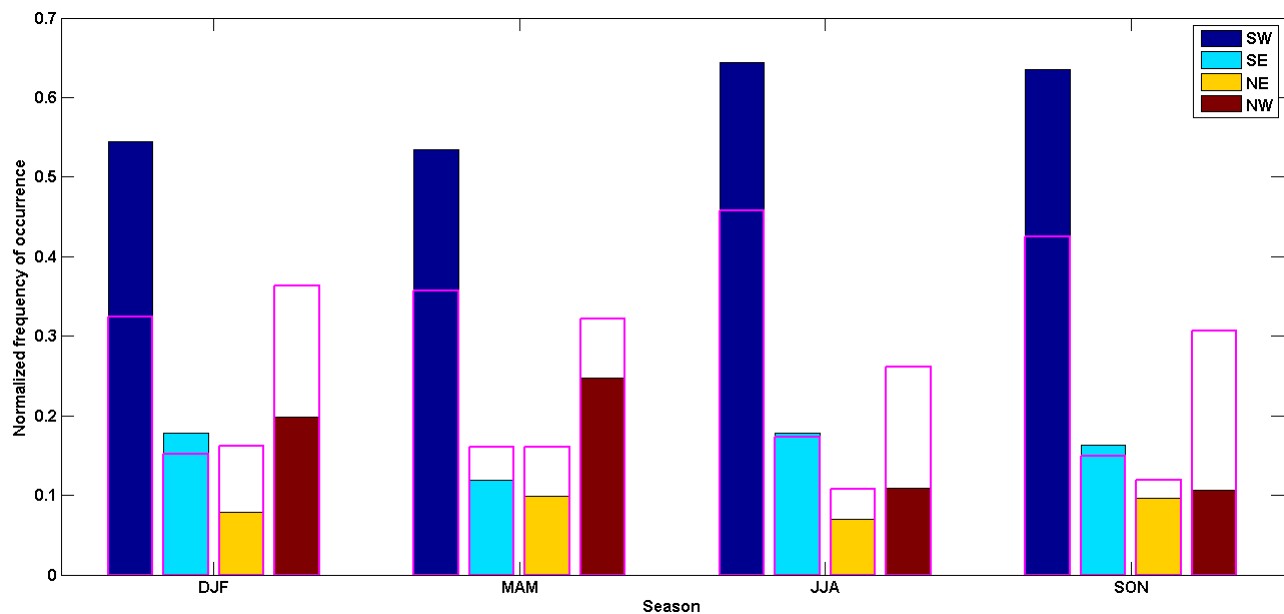

Fig. 4: Seasonal normalized frequency of occurrence of a particular wind direction at 850 hPa when NO$_2$ extreme pollution events were observed. The hollow magenta bars show normalized frequency under climatological conditions.

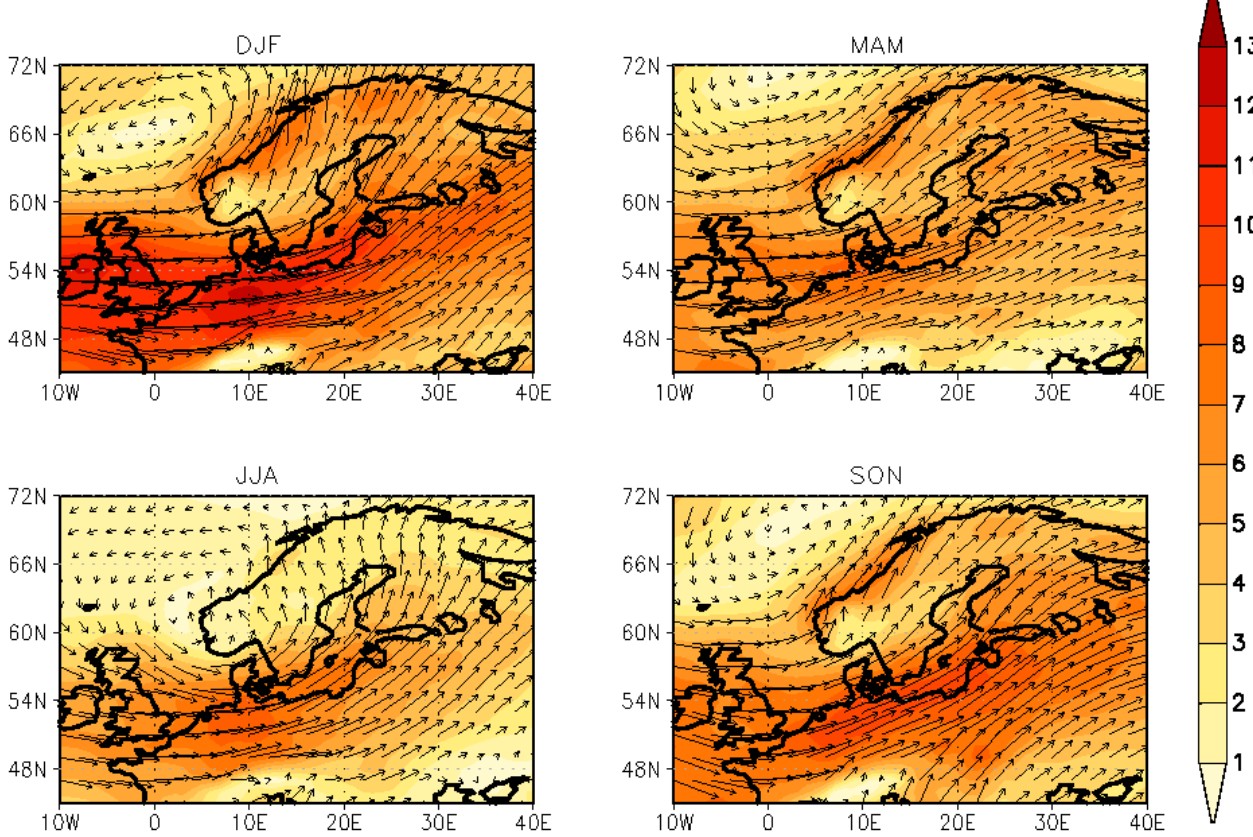





Fig. 5: Seasonal average wind strengths and direction at 850 hPa showing dominant circulation
pattern observed when NO$_2$ extreme pollution events occur.

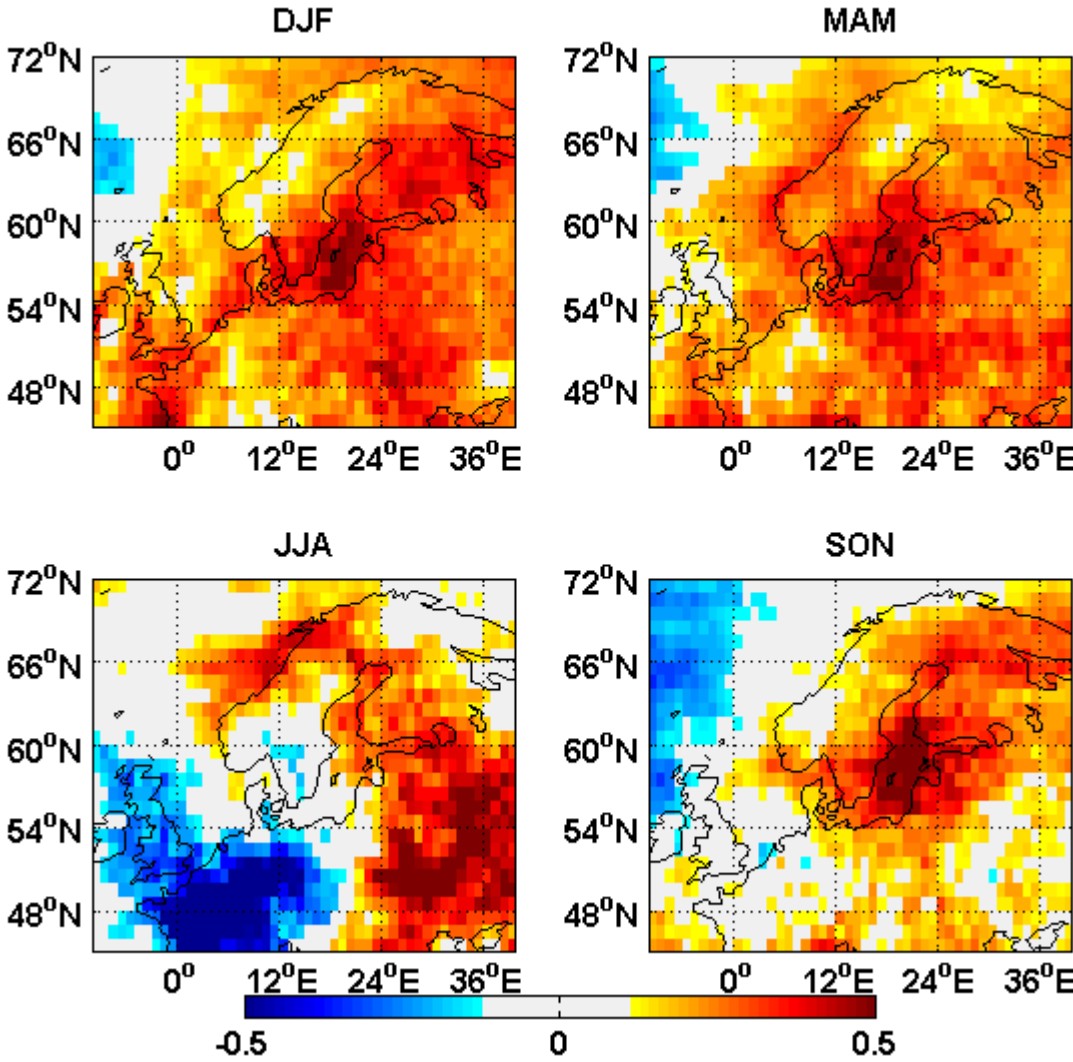

Fig. 6: The seasonal spatial patterns of specific humidity anomalies (g/kg) during extreme $NO_2$ pollution events.


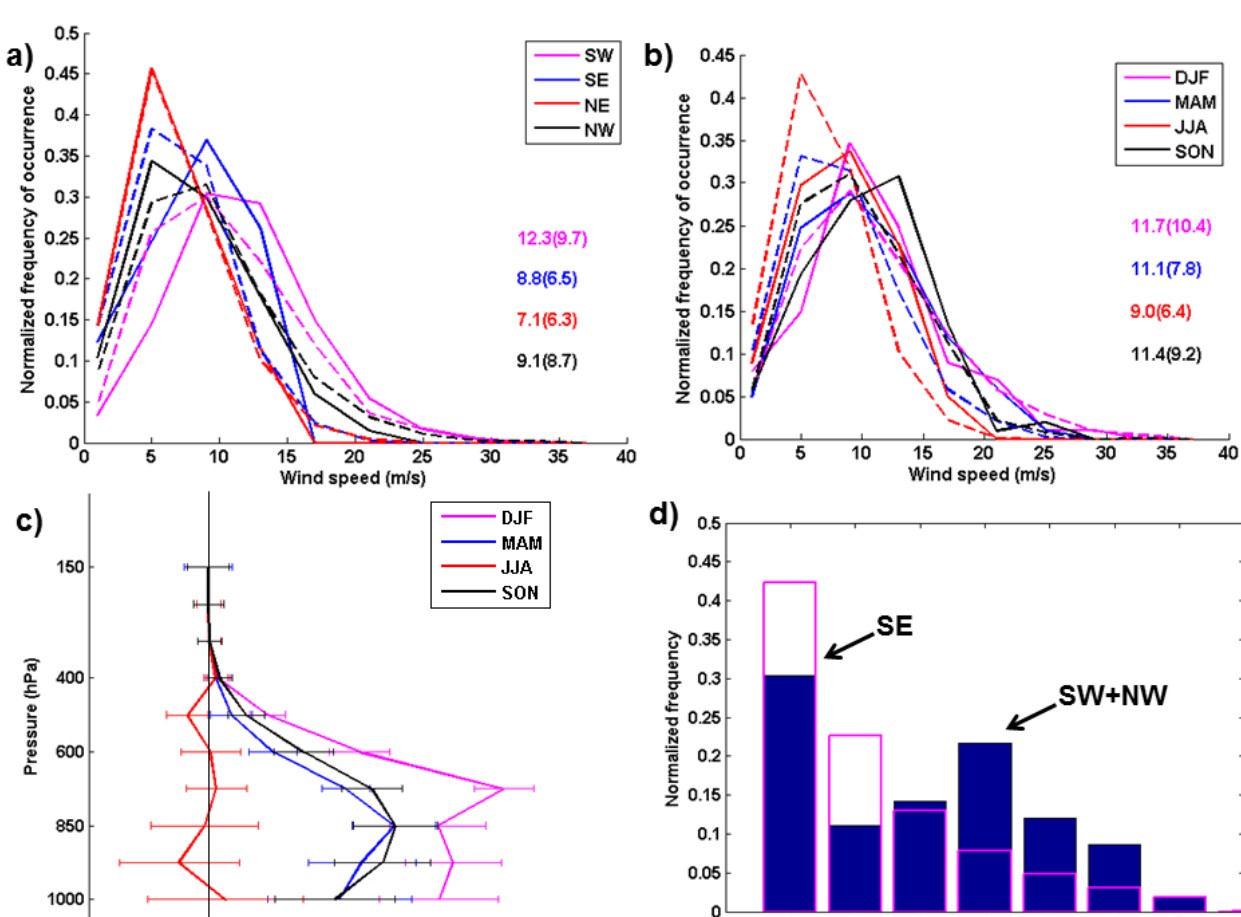



Fig. 7: a) Histograms of wind speeds (m/s) at 850 hPa over the center of the study area (55N-60N,
11E-20E) during extreme events (solid lines) and climatological conditions (dotted lines, 2004-
2015) when data are partitioned for different wind directions. The numbers show average wind
speeds (m/s) during extreme events and in brackets under climatological conditions.  b) Same as in
(a), but when wind data are partitioned for different seasons.  c) Vertical anomalies of specific
humidity (g/kg) during extreme events with horizontal bars showing standard deviations. d)
Persistency of wind directions as a function of number of continuous days. The magenta bars show
persistency under climatological conditions.







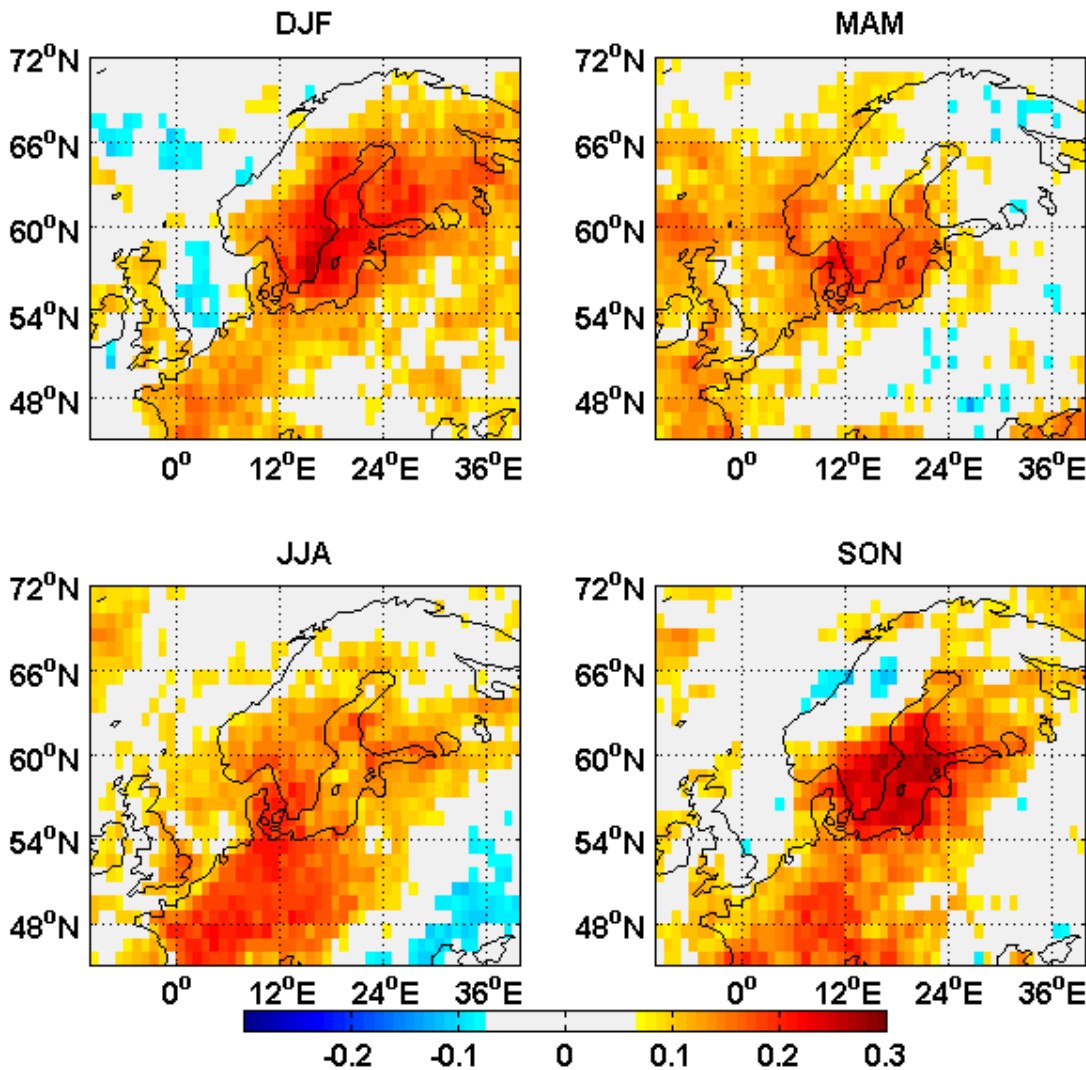



Fig. 8: Total cloud fraction anomalies observed during extreme events based on AIRS data.









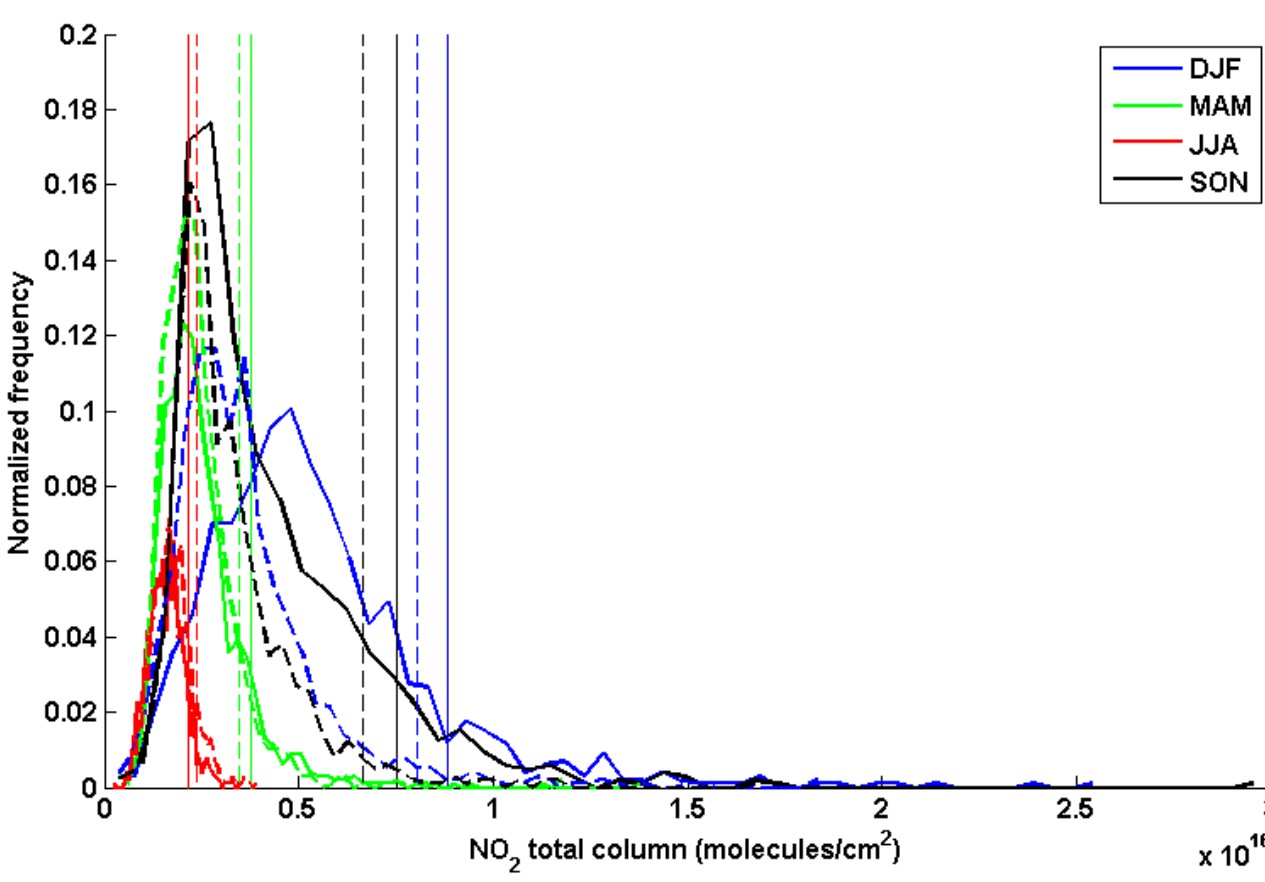



Fig. 9: Seasonal histograms of total column tropospheric NO$_2$ over the centre of the study area
(55N-60N, 11E-20E) and corresponding 90%ile thresholds (shown by vertical lines). The solid lines
show histograms based on retrievals under partially cloudy conditions, while the dotted lines show
histograms based only on cloud cleared retrievals.






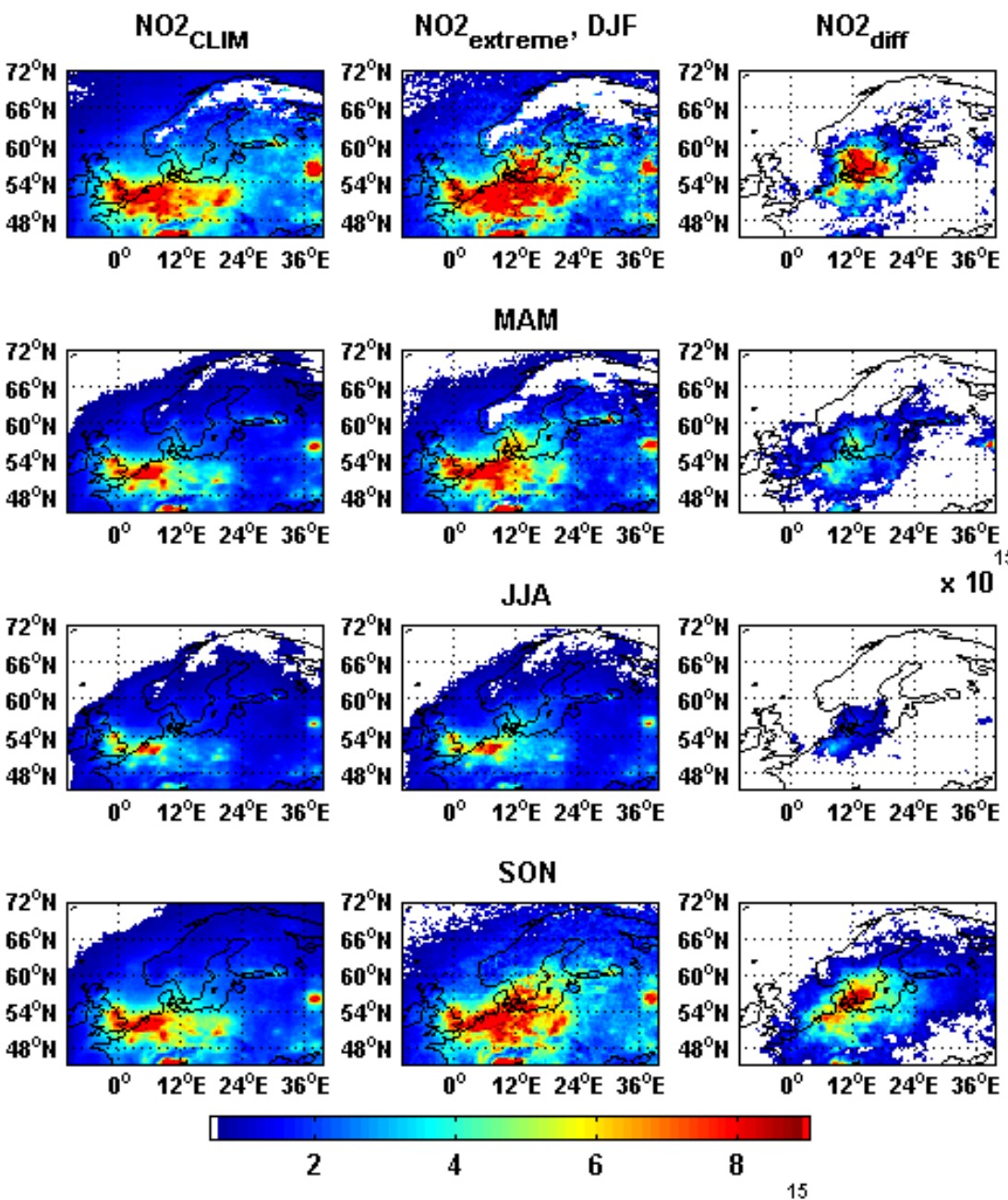


Fig. 10: Seasonal, climatological average tropospheric $NO_2$ total column (first column) based only
on cloud screened OMI data (2004-2015), $NO_2$ distribution during extreme events (second column,
also based on cloud screened data) and the difference between the two (third column). The units are
in molecules/cm$^2$.








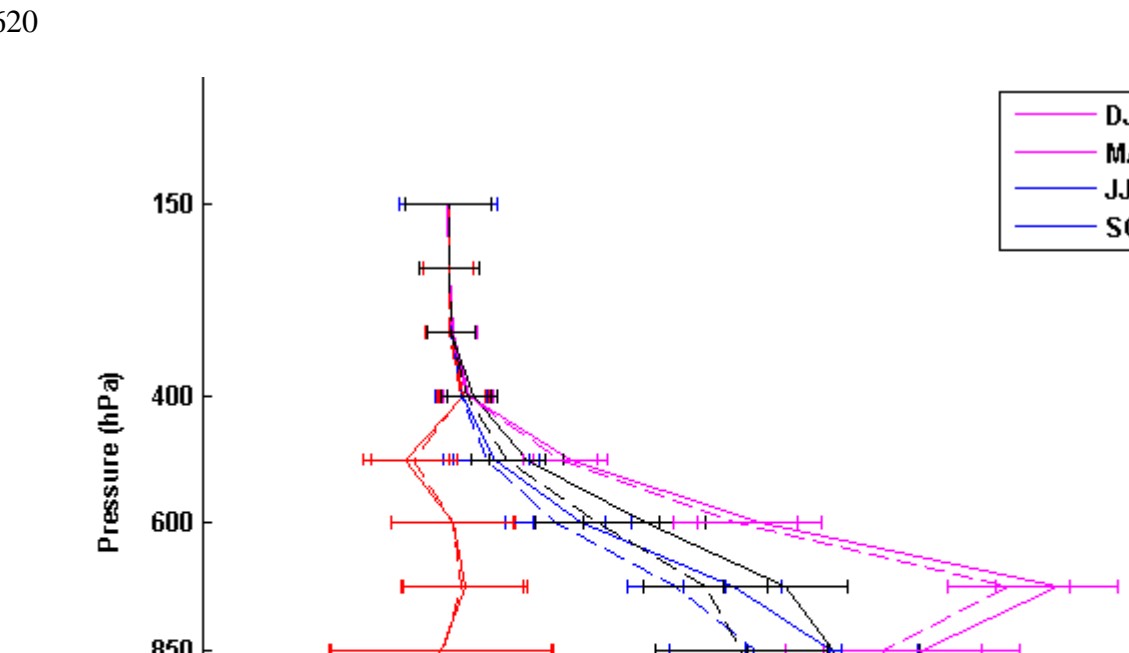



Fig. 11: Vertical anomalies of specific humidity (g/kg) during extreme events with horizontal bars
showing standard deviations. The solid lines show anomalies under partially cloudy retrievals and
dotted lines based on cloud screened retrievals.
