# Peer review of "1. Introduction"

_Atmospheric Chemistry and Physics, 2016_

## Referee Comment (RC1) · Anonymous Referee #1 · 14 Mar 2017

In this manuscript, Thomas and Devasthale present an analysis of the meteorological conditions associated with high NO2 concentrations over Scandinavia (as inferred by satellite NO2 columns from the OMI instrument). They find that the highest NO2 in each season is predominantly associated with south-westerly winds, and identify a clear 'transport pathway' using ERA reanalysis. The meteorological conditions tended to persist for either 1 day or 3-5 days, the latter potentially allowing for the longest range transport.

General Comments:

Overall, the paper is well-written with good consideration of related work. The methods are mostly well laid out, and the results are presented clearly. However, I have a couple

important concerns.

My main concern is that it is not clear what value has been added from this analysis. As the authors state, the dominance of south-westerly winds during "extreme" pollution is consistent with pollutant transport discussed elsewhere. What has been learned from this particular analysis of satellite NO2 columns in combination with ERA reanalysis data? What new concepts or ideas have been presented? In my opinion, in its current form the scientific significance of this paper is bordering on poor. For publication in ACP, the authors must make a stronger case for the novelty of their findings.

My secondary concern is that they tend to analyze the meteorological conditions during the "extreme" conditions only, without showing us that these conditions are unique to the "extremes" (and thus a defining factor in extreme pollutant concentrations for the region). In other words, the authors haven't clearly shown that the patterns associated with "extreme" NO2 are any different from patterns under more typical pollutant concentrations. What, then, is gained? The "clear transport pathway" could still exist under non-extreme conditions, in which case an explanation for extreme concentrations would need to look at other factors. I suggest the authors prove that these meteorological conditions tend to be specific to extreme NO2 by explicitly showing that the meteorological conditions for the other 90 percent of the time look different.

A more semantic concern: I am not a fan of their use of the term "extreme" throughout the manuscript. Given their definition of an "extreme" event (90th percentile concentrations in each individual season), an "extreme" event in July is nothing like an "extreme" event in January. The analysis is therefore a bit confusing in terms of potential relevance or impact. I wish they would use something a little more qualified throughout their manuscript. For example, could they say "the highest summertime concentrations" instead of "extreme NO2 concentrations".

Specific Comments:

Lines 39-41: Why not include soils amongst the other sources of NOx?

Line 43: One rarely hears NO2 referred to as a strong oxidizing agent with respect to atmospheric chemistry. I suggest rephrasing.

Line 52: I suggest the authors include a more recent projection, given the Lamarque et al. 2005 result is based on IPCC SRES A2 scenario which has very different NOx emission projects to 2100 from, say, the more recent RCP scenarios.

Line 100: Please provide a link to the direct source of the OMI data.

Line 102: "Under cloudy conditions": Please clarify that this means you have not applied any cloud fraction threshold for quality assurance. More importantly, I understand the motivation for this was to include more data (and avoid bias), but these observations result in very high uncertainty. Zien et al. (2014) go through quite a bit of detail explaining their unique treatment of cloudy data, and propose a new computation for the air mass factor calculation in these cases. However, if the authors of the present study are simply using the standard product retrieval (OMNO2d V.3), I expect the cloudy observations to have little realistic meaning. In the best case, over polluted regions with very low cloud fraction, the NO2 tropospheric retrieval still has between 35-60% error (Boersma et al. 2004). This will be much higher for cloudy data. While the authors have made a good case for including the observations, they have not discussed how the poor quality of such observations could impact their results. I.e., what good is a lot of data, if most of it is bad? This must be addressed.

Line 110: Please provide a link to the direct source of the AIRS data.

Line 113: Please provide a link to the direct source of the ERA reanalysis data.

Line 130: Please explicitly state here how the seasons were defined (DJF, MAM, JJA, SON).

Line 137: "It is interesting to see that...": I would argue that this is not at all interesting, but an obvious outcome of their design, given the definition of "extreme" and the seasonality of NO2.

[Figure]

Line 141: The authors note a "bimodal peak" in the distribution of NO2 extreme events. However, the definition of the seasons (DJF, MAM, JJA, SON) is extremely important here. As they have previously stated, NO2 has strong seasonality, peaking in the winter. So, if "spring" is defined as March-April-May, it is not surprising that most of the 90th percentile values for NO2 in this season will occur in March (the closest winter month). Likewise, if the fall is defined as Sept-Oct-Nov, it is obvious that most of the 90th percentile values for NO2 in this season would occur in November (the closest winter month). Thus, the "bimodal" shape of Figure 2b is almost certainly an artifact of their experimental design. I therefore don't understand what we learn from this figure, and suggest removing it, and any discussion of it.

Line 183: Are the specific humidity anomalies calculated with respect to a long-term mean? Or with respect to the other 90% of the NO2 concentration days? Please define how the "anomaly" has been calculated (this would be most appropriate in the Methods section).

Line 203: "higher wind speeds.. during extreme events compared to climatological conditions": Where are the climatological conditions shown? As I mention above, I suggest including figures of the climatological meteorology in all cases, for a more obvious comparison to the "extreme" conditions.

Line 209: If I wanted to reproduce this data, how was "persistency" actually deter- mined? I.e. what computational steps were performed on the meteorological data to evaluate this.

---

## Referee Comment (RC2) · Anonymous Referee #2 · 7 Apr 2017

In their manuscript "Typical meteorological conditions associated with extreme nitrogen dioxide (NO2) pollution events over Scandinavia", Thomas and Devasthale report on a study evaluating the meteorological conditions under which the highest tropospheric NO2 columns are observed in OMI data over Scandinavia. Their results show, that such events are linked to situations in which transport from the polluted regions in Europe towards Scandinavia occurs, that such events are mostly observed in winter and spring and that they persist for several days.

The topic of the study (impact of meteorology and long-range transport on pollution) is interesting and fits well into ACP. The paper is also well written, clearly structured and to the point. I have however several concerns with respect to the relevance and

also the methodology of the study which need to be addressed before the paper can be considered for publication.

**General points**

- Probably the most important point is that I'm not sure what the relevance of the results presented in this manuscript really is. It is not surprising that pollution transport from Germany and the Benelux countries impacts on Southern Scandinavian air quality. As there is no attempt made to quantify the impact in absolute or relative terms, the study does little more than confirming what one would have guessed anyways. I think it would be good to try to become more quantitative in the sense of how many days are affected, what are the mean and maximum anomalies, and what is the relation to pollution from local sources.

- A second very important point is the use of OMI satellite data without separating cloud free and cloudy situations. While the argument for this approach is clear as many transport events are associated to clouds, such data cannot be easily interpreted as for cloudy conditions, the assumptions made in the retrieval become very important for the results. In the current manuscript, this point is not addressed at all and I think the authors need to investigate differences between cloudy and clear sky averages in order to better understand the impact of clouds on the satellite data. They also need to discuss uncertainties linked to the satellite retrievals.

- The tacit assumption made in this study is that NO2 observed from satellite (partly above clouds) is indicative of enhanced NO2 levels on the ground. I'm not convinced that this is always the case during transport events and it would be good to support the timing and location of their extreme NO2 events by at least some surface observations showing that in deed air quality on the ground was also poor during the satellite observed pollution events.

- Definition of extreme cases is another critical aspect and I think that given the large seasonality of NO2, monthly thresholds would be better than seasonal thresholds. I'm also a bit confused by the relevance of Figure 2b) showing the number of extreme events per month – isn't the number of extreme events per season constant the ways the authors define their thresholds, and therefore the distribution over months just reflecting the seasonality of NO2?

- Considering Figure 3, I'm wondering why the situations with higher NO2 over Southern Scandinavia appear to also have higher than normal NO2 over the supposed source regions. Does this mean that under these conditions, pollution is accumulating in general? If this would be simple transport from Central Europe to Scandinavia, I would expect to see less NO2 in the source region or what am I missing here?

**Minor points**

- Line 40: Add soil emissions

- Line 47: Does NO2 really affect psychological health?

- Figure 3: Are these total or tropospheric columns?

- Figure 6: Not sure what is "the same as in Fig. 5" here

- Figure 7 a / b are difficult to read (too many lines)

- While the article is overall well written, there are many places in which I would add / remove articles. I therefore recommend another round of proof reading to fix these and other small English problems.

---

## Author Comment (AC1) · 31 May 2017

**Response to Referee #1**

We thank the referee for constructive comments that lead to substantial improvements in the revised manuscript. Based on comments by both referees, a major revision of the manuscript is carried out. In particular we have tried to address the following three concerns.

1) The entire analysis is revised to investigate the sensitivity of our results to using OMI retrievals under cloud-cleared versus cloudy conditions. The analysis of AIRS and reanalysis datasets is also revised accordingly.
2) The revised analysis is now based on individual months (instead of seasons) to take into account even the monthly variability.
3) A stronger case is made to clarify precise contribution of the present work.

Please find below point-by-point reply to your general and specific comments.

In this manuscript, Thomas and Devasthale present an analysis of the meteorological conditions associated with high NO2 concentrations over Scandinavia (as inferred by satellite NO2 columns from the OMI instrument). They find that the highest NO2 in each season is predominantly associated with south-westerly winds, and identify a clear 'transport pathway' using ERA reanalysis. The meteorological conditions tended to persist for either 1 day or 3-5 days, the latter potentially allowing for the longest range transport.

General Comments:

Overall, the paper is well-written with good consideration of related work. The methods are mostly well laid out, and the results are presented clearly. However, I have a couple important concerns.

We thank the referee for encouraging comments.

My main concern is that it is not clear what value has been added from this analysis. As the authors state, the dominance of south-westerly winds during "extreme" pollution is consistent with pollutant transport discussed elsewhere. What has been learned from this particular analysis of satellite NO2 columns in combination with ERA reanalysis data? What new concepts or ideas have been presented? In my opinion, in its current form the scientific significance of this paper is bordering on poor. For publication in ACP, the authors must make a stronger case for the novelty of their findings.

We agree that we had not made our contribution clear enough for readers. Below we would briefly like to point out our value addition.

1) First of all, we are not aware of such detailed, observationally constrained analysis of weather patterns associated with hundreds of extreme NO2 pollution events over Scandinavia, especially by making synergistic use of satellite and reanalysis datasets.
2) We have quantified relative importance of different wind directions during these events in various seasons.
3) We have also investigated the persistency of weather patterns during extreme events and which wind directions explain the observed modes in the persistency.

All of these results and statistics presented here will be helpful for chemistry and tracer transport models while studying the impact of norward long-range transport of pollutants.

Hopefully, the revised text will make it clear for the reader.

My secondary concern is that they tend to analyze the meteorological conditions during the "extreme" conditions only, without showing us that these conditions are unique to the "extremes" (and thus a defining factor in extreme pollutant concentrations for the region). In other words, the authors haven't clearly shown that the patterns associated with "extreme" NO2 are any different from patterns under more typical pollutant concentrations. What, then, is gained? The "clear transport pathway" could still exist under non-extreme conditions, in which case an explanation for extreme concentrations would need to look at other factors. I suggest the authors prove that these meteorological conditions tend to be specific to extreme NO2 by explicitly showing that the meteorological conditions for the other 90 percent of the time look different.

The aim of the study was to analyse meteorological conditions during extreme pollution events, *irrespective* of the fact whether these conditions turn out to be different or not compared to climatology. We respectfully disagree that nothing is gained if they don't turn out to be very different. It is in fact equally important to point out how challenging it will be for chemistry and tracer transport models to disentangle typical patterns associated with extreme events.

A more semantic concern: I am not a fan of their use of the term "extreme" throughout the manuscript. Given their definition of an "extreme" event (90th percentile concentrations in each individual season), an "extreme" event in July is nothing like an "extreme" event in January. The analysis is therefore a bit confusing in terms of potential relevance or impact. I wish they would use something a little more qualified throughout their manuscript. For example, could they say "the highest summertime concentrations" instead of "extreme NO2 concentrations".

Since the revised analysis is based on applying thresholds on individual months, the chosen extreme event is representative of that particular month. We appreciate the fact that an extreme event in Jan is different than in July and that is why we have chosen percentile matric instead of a fixed threshold on the absolute values of NO2. We believe such 90%ile threshold will allow the investigation of meteorological conditions in a fair manner during extreme events.

Specific Comments:
Lines 39-41: Why not include soils amongst the other sources of Nox?

Included.

Line 43: One rarely hears NO2 referred to as a strong oxidizing agent with respect to atmospheric chemistry. I suggest rephrasing.

This line is rephrased.

Line 52: I suggest the authors include a more recent projection, given the Lamarque et al. 2005 result is based on IPCC SRES A2 scenario which has very different Nox emission projects to 2100 from, say, the more recent RCP scenarios.

Included.

Line 100: Please provide a link to the direct source of the OMI data.

The OMI data were obtained from NASA's GES DISC. The following link is included in the revised version.
https://disc.gsfc.nasa.gov/Aura/data-holdings/OMI/omno2d_v003.shtml

Line 102: "Under cloudy conditions": Please clarify that this means you have not applied any cloud fraction threshold for quality assurance. More importantly, I understand the motivation for this was to include more data (and avoid bias), but these observations result in very high uncertainty. Zien et al. (2014) go through quite a bit of detail explaining their unique treatment of cloudy data, and propose a new computation for the air mass factor calculation in these cases. However, if the authors of the present study are simply using the standard product retrieval (OMNO2d V.3), I expect the cloudy observations to have little realistic meaning. In the best case, over polluted regions with very low cloud fraction, the NO2 tropospheric retrieval still has between 35-60% error (Boersma et al. 2004). This will be much higher for cloudy data. While the authors have made a good case for including the observations, they have not discussed how the poor quality of such observations could impact their results. I.e., what good is a lot of data, if most of it is bad? This must be addressed.

This is indeed a good point. However, purely from the user point of view, we find it difficult to see how we can improve our current analysis of satellite data. We have done our best to follow recommendations for the users. We appreciate that even the clear-sky retrievals could have large error bars, but based on the sensitivity analysis (please see response to another reviewer) and the consistency of spatial distribution of NO2 during all-sky and clear-sky conditions, we are confident that there aren't major artefacts affecting our analysis of OMI data.

Line 110: Please provide a link to the direct source of the AIRS data.

The AIRS daily gridded data (AIRX3STD) were also obtained from the NASA's GES DISC AIRS Holding. The following link is provided in the revised version.
https://disc.gsfc.nasa.gov/uui/search/%22AIRS%22

Line 113: Please provide a link to the direct source of the ERA reanalysis data.

ERA reanalysis data were directly downloaded from the ECMWF's data server.

http://apps.ecmwf.int/datasets/data/interim-full-daily/levtype=sfc/

Line 130: Please explicitly state here how the seasons were defined (DJF, MAM, JJA,SON).

This is now clarified. Please note that while the revised analysis is based on individual months, the results are presented for seasons to avoid too many smaller subplots.

Line 137: "It is interesting to see that...": I would argue that this is not at all interesting, but an obvious outcome of their design, given the definition of "extreme" and theseasonality of NO2.

Here we merely wanted to point out that, due to long-tailed distribution of NO2 in DJF, the chosen threshold for the winter months is higher than the one in the summer months.

Line 141: The authors note a "bimodal peak" in the distribution of NO2 extreme events. However, the definition of the seasons (DJF, MAM, JJA, SON) is extremely important here. As they have previously stated, NO2 has strong seasonality, peaking in the winter. So, if "spring" is defined as March-April-May, it is not surprising that most of the 90th percentile values for NO2 in this season will occur in March (the closest winter month). Likewise, if the fall is defined as Sept-Oct-Nov, it is obvious that most of the 90th percentile values for NO2 in this season would occur in November (the closest winter month). Thus, the "bimodal" shape of Figure 2b is almost certainly an artifact of their experimental design. I therefore don't understand what we learn from this figure, and suggest removing it, and any discussion of it.

The Fig. 2b is dropped in the revised manuscript.

Line 183: Are the specific humidity anomalies calculated with respect to a long-term mean? Or with respect to the other 90% of the NO2 concentration days? Please define how the "anomaly" has been calculated (this would be most appropriate in the Methods section).

Yes, the specific humidity anomalies are computed with respect to a long-term mean. This climatological mean is subtracted from the average humidity anomalies observed during extreme events.

Line 203: "higher wind speeds.. during extreme events compared to climatological conditions": Where are the climatological conditions shown? As I mention above, I suggest including figures of the climatological meteorology in all cases, for a more obvious comparison to the "extreme" conditions.

Please note that Figs. 7a and 7b also show wind distribution under climatological conditions (dotted lines). Furthermore, the numbers in brackets show average wind speeds under climatological conditions as a reference.

Line 209: If I wanted to reproduce this data, how was "persistency" actually determined? I.e. what computational steps were performed on the meteorological data to evaluate this.

The persistency is defined as follows. If an extreme event is observed on any particular day, the wind strength and wind direction is computed from the u and v vectors. We then check how many days back in time this particular wind direction is sustained over the centre of the study area.

---

## Author Comment (AC2) · 31 May 2017

**Response to Referee #2**

We thank the referee for constructive comments that lead to substantial improvements in the revised manuscript. Based on comments by both referees, a major revision of the manuscript is carried out. In particular we have tried to address the following three concerns.

1) The entire analysis is revised to investigate the sensitivity of our results to using OMI retrievals under cloud-cleared versus cloudy conditions. The analysis of AIRS and reanalysis datasets is also revised accordingly.
2) The revised analysis is now based on individual months (instead of seasons) to take into account even the monthly variability.
3) A stronger case is made to clarify precise contribution of the present work.

Please find below point-by-point reply to your general and specific comments.

In their manuscript "Typical meteorological conditions associated with extreme nitrogen dioxide (NO2) pollution events over Scandinavia", Thomas and Devasthale report on a study evaluating the meteorological conditions under which the highest tropospheric NO2 columns are observed in OMI data over Scandinavia. Their results show, that such events are linked to situations in which transport from the polluted regions in Europe towards Scandinavia occurs, that such events are mostly observed in winter and spring and that they persist for several days. The topic of the study (impact of meteorology and long-range transport on pollution) is interesting and fits well into ACP. The paper is also well written, clearly structured and to the point.

We thank the referee for encouraging comments.

I have however several concerns with respect to the relevance and also the methodology of the study which need to be addressed before the paper can be considered for publication.

General points
• Probably the most important point is that I'm not sure what the relevance of the results presented in this manuscript really is. It is not surprising that pollution transport from Germany and the Benelux countries impacts on Southern Scandinavian air quality. As there is no attempt made to quantify the impact in absolute or relative terms, the study does little more than confirming what one would have guessed anyways. I think it would be good to try to become more quantitative in the sense of how many days are affected, what are the mean and maximum anomalies, and what is the relation to pollution from local sources.

Please note that the main aim of the study is to quantify typical meteorological conditions during the extreme pollution events and not to show that the transport from Germany and the Benelux countries impacts on Southern Scandinavian air quality. The latter conclusion, while still interesting since it is based on observationally constrained datasets, is the bi-product from the inferral of the combined analysis of circulation patterns and changes in humidity and cloudiness. The emphasis is rather given on describing meteorological conditions and whether and how they differ from the climatological conditions.

• A second very important point is the use of OMI satellite data without separating cloud free and cloudy situations. While the argument for this approach is clear as many transport events are associated to clouds, such data cannot be easily interpreted as for cloudy conditions, the assumptions made in the retrieval become very important for the results. In the current manuscript, this point is not addressed at all and I think the authors need to investigate differences between cloudy and clear sky averages in order to better understand the impact of clouds on the satellite data. They also need to discuss uncertainties linked to the satellite retrievals.

This is indeed a good point, which is also raised by another referee. We have now carried out sensitivity analysis to check to what extent our selection of extreme events, the spatial distribution of NO2 and corresponding humidity anomalies are affected by restricting to only clear-sky conditions.

- A figure below shows the spatial distribution of NO2 during only clear-sky conditions for climatological conditions, during extreme events and the differences between the two. This figure can be compared to Fig. 3 in the manuscript. It can be seen that the general distribution of NO2 and its anomalies in the figure below are quite similar to Fig. 3.

[Figure]

- A figure below shows a comparison of the NO2 histograms over the center of the study area. The solid lines show NO2 distribution under all-sky conditions and dotted lines under clear-sky. The differences in the summer half year are minimal. The differences in the winter half year are not large enough to change either the spatial distribution of NO2 as shown above or to affect the tendencies of specific humidity anomalies as shown below.

[Figure]

- We further investigated the vertical structure of specific humidity anomalies under clear-sky conditions. A figure below shows the comparison thereof. Once again it is seen that the general tendency that humidity is increased during the extreme events does not change even under the clear-sky conditions.

[Figure]

These results provide us confidence that our analysis is most likely not affected by the contamination from clouds.

• The tacit assumption made in this study is that NO2 observed from satellite (partly above clouds) is indicative of enhanced NO2 levels on the ground. I'm not convinced that this is always the case during transport events and it would be good to support the timing and location of their extreme NO2 events by at least some surface observations showing that in deed air quality on the ground was also poor during the satellite observed pollution events.

This is also a good point, but we have to admit that addressing this would be out of the scope of the present study. As mere users of satellite data sets we have to put faith in these data sets (and the work done by the respective Science Teams) and hope that the tropospheric columns would capture the variability and coupling between the concentrations in the free troposphere and near-ground. If the referee insists and the extra time is granted by the editor, we would be happy to cross-check with surface observations. We also kindly request the referee if he/she would point out to us relevant references that discuss such discrepancy.

Definition of extreme cases is another critical aspect and I think that given the large seasonality of NO2, monthly thresholds would be better than seasonal thresholds. I'm also a bit confused by the relevance of Figure 2b) showing the number of extreme events per month – isn't the number of extreme events per season constant the ways the authors define their thresholds, and therefore the distribution over months just reflecting the seasonality of NO2?

In the updated version of the manuscript, Fig. 2b is dropped as this was also demanded by the other referee. We have further revised the entire analysis and based it on the monthly instead of seasonal thresholds. Please note that the major conclusions do not change even when we select extreme events based on the monthly thresholds.

• Considering Figure 3, I'm wondering why the situations with higher NO2 over Southern Scandinavia appear to also have higher than normal NO2 over the supposed source regions. Does this mean that under these conditions, pollution is accumulating in general? If this would be simple transport from Central Europe to Scandinavia, I would expect to see less NO2 in the source region or what am I missing here?

Please note that to qualify for being an extreme event over Southern Scandinavia, the pollutant levels in the source regions also need to be higher than usual together with favourable circulation patterns.

Minor points
• Line 40: Add soil emissions

Added.

• Line 47: Does NO2 really affect psychological health?

It has been claimed that it does, but indirectly by affecting physical health and thereby psychological. In the

revised version, the reference to psychological health is removed.

• Figure 3: Are these total or tropospheric columns?

Only tropospheric columns are analysed. It is now clarified in the revised manuscript.

• Figure 6: Not sure what is "the same as in Fig. 5" here

This is clearly a mistake. Thanks for pointing it out. The Figure 6 shows the specific humidity anomalies (averages during extreme events minus climatological conditions).

• Figure 7 a / b are difficult to read (too many lines)

We agree that there are too many lines in those figures, but it was necessary to do so to show climatological conditions. For a quick and easy reference to the reader, the average values of wind speeds are also shown in those subplots.

• While the article is overall well written, there are many places in which I would add / remove articles. I therefore recommend another round of proof reading to fix these and other small English problems.

We have tried to fix misplacing of articles in the revised version.

---

## Author Response (AR2)

Response to Referee #1

We are once again thankful to the referee for her/his constructive comments. Please find below point by point response.

In response to my reviewer comment, the authors added a definition of "persistency" in their analysis which is very helpful. Still, I find the definition lacking in technical details. The authors state that "it is checked how many days back in time that particular wind direction was continuously sustained". This must involve some threshold of acceptable deviation. Were winds allowed to deviate by up to 10 degrees? 50 degrees? 1 degree? or did they truly need to be continuously sustained, to the same decimal point? This is of course a trivial detail, but I don't understand why it can't be included in their manuscript to help readers understand their exact approach. And how sensitive is their identification of "two distinct modes" to whatever threshold they chose?

The following text is added in the revised version to further clarify the definition of persistency.

"If an extreme event is observed, the wind speed and wind direction are computed for the last 10 days. It is then checked how many days back in time that particular wind direction was *continuously* sustained and that wind direction is not changed by more than $\pm15^0$ (a third of the quadrant) during that time period. It is to be noted that the choice of the $\pm15^0$ threshold is based on the visual inspection of about 25 test cases. It was found that if a stricter threshold is used (requiring wind direction deviations less than $\pm5^0$) the sampling is considerably reduced for long persistency events. On the other hand, if a more relax threshold is used (allowing deviations up to $\pm30^0$) we incorporate tail ends of the events that persisted over neighbouring areas."

The authors have added a sensitivity test based on cloud screening which is certainly helpful. Still, in their manuscript methods they state "we allowed retrievals under partially cloudy conditions to be analyzed" (this is repeated in Section 4). This statement is misleading, since it implies to me that there is some filtering that has been done ("partially cloudy"). Surely some scenes may be fully cloudy. The authors need to state explicitly that they have not applied any filter for cloudy data, and that they test the impact of this later.

The word "partially" is causing confusion here and hence it is removed from the text, now implying that all cases (irrespective of partly or fully cloudy conditions) were analysed.

I actually agree with Reviewer 2 regarding the question about the assumption that NO2 observed from satellite is indicative of enhanced NO2 levels on the ground. But I also agree with the authors that this is not necessarily within the scope of their current paper. However, I am a little disturbed that the author response is that they "put faith" in the datasets and "hope" the tropospheric columns would capture the variability near the ground. It is not the responsibility of the satellite retrieval science teams to guarantee the tropospheric columns have any relevance to surface conditions. It is also not the responsibility of the referee to point to references that discuss a discrepancy, as requested by the authors in response. Rather, in my opinion, it is the role of the authors to convince us that there isn't a discrepancy, or argue that any discrepancy is not important to their conclusions (which may very well be the case). Perhaps a compromise would be for the authors to include a few comments/caveats to this point specifically, or to include references to other literature that supports any relevant assumptions.

We apologize for the poor choice of words and for not conveying the message properly. We do fully appreciate that in the end it is our responsibility to make sure that we use dataset properly for our purpose. The only point we wanted to make was that, as a user, we have gone through all relevant data documents and have tried to ensure that we use the data correctly. Since we are using retrievals only to select extreme events (rather than doing full scale transport analysis using retrievals) we thought our data handling should serve the purpose and that any validation work would be out of the scope of the present study.

We agree with the recent reviewer comment that the said discrepancy is not directly important for our work. This is because while characterizing meteorological conditions, we are interested in the enhanced $NO_2$ levels in the troposphere as a whole, not necessarily confined to the near-surface. In fact our study region could just be a part of the longer transit pathway for the eventual long-range transport of pollutants to the Arctic.

I commend the authors' approach to Figure 2, using monthly thresholds instead of seasonal thresholds. However, this figure is difficult to read. This could be corrected by simply including the absolute values of the 90th percentile for each month beside the legend labels. Also, why not label the colors by the month name, instead of a number (the rest of the plots refer to month names ("DJF", "SON", etc.)).

Figure 2 is revised. The months are labelled with names instead of numbers and corresponding 90%ile thresholds are also added in brackets.

**Response to Referee #2**

We are once again thankful to the referee for her/his constructive comments. Please find below point by point response.

The newly added sentences on page 11, line 51 do not look right to me - emissions are not given in ppb. Please check

Thanks. It is now corrected.

* if I have not overlooked this information, the threshold for the cloud screening which was applied in the test case is not given. Please add.

The following sentence is now added to clarify it.

"We required that cloud fraction is less than 10% in AIRS data and valid retrievals of OMI cloud cleared tropospheric column $NO_2$ are available."

* it would be really good to make Fig. 10 identical to Fig. 3 with respect to figure sizes and legend

Corrected.

* I do not understand the sentence on page 14, line 163 "By definition, NO2 anomalies during extreme events are similar in magnitude to climatological values over Scandinavia - please explain

Please note that we are referring to the anomalies (and not the absolute values).

* page 14, line 153: concentrations => columns

Corrected.

* There still are many small English issues which should be fixed before publication

We have tried to correct grammatical issues (in particular the use of articles).

[revised manuscript text omitted]